# BK Polyomavirus-Associated Nephropathy and Hemorrhagic Cystitis in Transplant Recipients—What We Understand and What Remains Unclear

**DOI:** 10.3390/v17091256

**Published:** 2025-09-17

**Authors:** Tang-Her Jaing, Yi-Lun Wang, Tsung-Yen Chang

**Affiliations:** Division of Hematology and Oncology, Department of Pediatrics, Chang Gung Memorial Hospital, Taoyuan 33315, China; g987669@gmail.com (Y.-L.W.); gisborne@cgmh.org.tw (T.-Y.C.)

**Keywords:** BK virus, hemorrhagic cystitis, allogeneic hematopoietic cell transplantation, pediatric

## Abstract

The reactivation of BK polyomavirus (BKPyV) during severe immunosuppression plays a crucial role in two significant syndromes observed in transplant recipients: BK polyomavirus-associated nephropathy (BKPyVAN) in kidney transplant patients and BK polyomavirus-associated hemorrhagic cystitis (BKPyV-HC) in hematopoietic cell transplant (HCT) recipients. This review aims to summarize the current understanding and lingering ambiguity by looking at three primary questions: (1) In cases with BKPyV-related illnesses in transplant patients, which diagnostic methods have the best track record of accuracy and success? (2) Which therapy approaches have the best track records of safety and efficacy in real-world clinical settings? (3) What can immunological research teach us about the development of future tailored treatments? Diagnosis involves the patient’s appearance, ruling out other potential causes, and employing quantitative PCR to identify active viral replication in urine or plasma. BKPyV-HC can vary from self-limited hematuria to potentially fatal bleeding, while BKPyVAN may lead to loss and dysfunction of the allograft. Reducing immunosuppression remains the key aspect of treatment. However, the effectiveness of antivirals (such cidofovir and leflunomide) is not always the same, and supporting measures depend on the syndrome. Researchers are looking into new immunotherapies, such as virus-specific cytotoxic T cells. Due to the intricate viro-immunopathology and lack of defined treatment regimens, future initiatives should focus on prospective studies to establish validated thresholds, enhance management algorithms, and integrate immune surveillance into individualized therapy.

## 1. Introduction

### 1.1. BK Virus Overview

BK polyomavirus (BKPyV), a member of the *Betapolyomavirus* genus within the *Polyomaviridae* family, is a ubiquitous virus implicated in several clinically significant conditions, particularly in immunocompromised individuals [1]. First identified in the 1970s alongside JC polyomavirus (JCPyV), BKPyV was among the earliest human polyomaviruses discovered following the isolation of murine polyomavirus (MPyV) in 1953, which had oncogenic potential in neonatal mice [2]. As of now, over a dozen human polyomaviruses have been identified, with the majority uncovered in the last twenty years [3], including the KI and WU polyomaviruses, which were identified in 2007 [4]. Seroprevalence studies reveal that BKPyV infects a significant portion of the population in early childhood, leading to lifelong latency, predominantly in the urogenital epithelium [5].

BKPyV reactivation in the context of immunosuppression—such as in kidney transplantation or allogeneic hematopoietic cell transplantation (HCT)—can result in severe complications, notably BKPyV-associated nephropathy (BKPyVN) and BK virus-associated hemorrhagic cystitis (BKPyV-HC). BKPyV-HC remains a major complication in HCT recipients, contributing to increased morbidity, prolonged hospitalization, and greater healthcare resource utilization. Currently, there are no standardized preventive or therapeutic protocols despite its clinical importance. Existing treatment approaches, such as immunosuppression modulation, antiviral agents (e.g., cidofovir, leflunomide), and supportive care, show inconsistent efficacy [6].

This review offers an updated overview of BKPyV-HC management strategies, focusing on recent advances, including novel immunotherapeutic approaches and biomarker-guided interventions that have demonstrated promising preliminary outcomes in recent clinical studies.

### 1.2. Epidemiology and Risk Factors for BKPyV-HC

BKPyV is a ubiquitous, non-enveloped, double-stranded DNA virus that establishes lifelong latency in the urinary tract. In immunocompromised hosts, particularly HCT recipients, viral reactivation can lead to BKPyV-HC.

The incidence of BKPyV-HC in allogeneic HCT recipients varies significantly, from about 5% to 25%, influenced by factors like conditioning regimen intensity, graft source, and GVHD prophylaxis strategy [7,8]. Higher rates are notably observed in recipients undergoing myeloablative conditioning and those receiving haploidentical transplants, especially with post-transplant cyclophosphamide (PTCy) for GVHD prophylaxis [9].

Recent multicenter studies have shown an increasing burden of BKPyV-HC in expanding haploidentical HCT programs, indicating both greater utilization and related immunologic challenges [10]. Geographic variability in incidence has been reported, but data are limited due to inconsistent screening practices and diagnostic criteria.

Due to these variations, reliable epidemiologic data remain heterogeneous, and most estimates are derived from retrospective or single-institution studies. Ongoing efforts are focused on establishing consistent diagnostic cutoffs and prospective surveillance strategies to better quantify and compare the incidence across transplant populations [11,12].

## 2. Literature Review Methodology

A narrative literature review was conducted to examine BKPyV in transplant recipients, following the PRISMA 2020 guidelines [13]. A thorough literature search was conducted in PubMed and Google Scholar for studies published from January 2000 to April 2025. Only English-language studies involving human subjects were included in the search. Eligible studies included original research articles, clinical trials, and relevant observational studies that focused on the diagnosis, treatment, or clinical outcomes of BKPyVN and BKPyV-HC in renal transplant or HCT recipients. Case reports, review articles, unrelated studies, and publications lacking full-text access were excluded.

After deduplication, titles and abstracts were screened independently by two reviewers for relevance. Full-text review of potentially eligible articles was likewise conducted in duplicate to confirm final inclusion. Discrepancies at either screening stage were resolved through consensus with a third reviewer. Data on study design, patient characteristics, diagnostic criteria, treatment strategies, and outcomes were extracted in a standardized format and synthesized qualitatively due to anticipated study heterogeneity.

## 3. Pathogenicity of BK Virus on the Urinary System

BKPyV primarily exhibits tropism for uroepithelial cells within the urinary tract. Following an initial asymptomatic infection during childhood, the virus establishes persistent latency in the epithelial cells of the renal tubules and bladder [14]. Under conditions of immunosuppression—such as in HCT or kidney transplantation—BKPyV can reactivate, leading to a spectrum of urologic complications [15].

### 3.1. Cellular Tropism and Reactivation

BKPyV preferentially targets renal tubular epithelial cells and bladder urothelium. Reactivation occurs due to compromised cellular immunity, especially reduced BKPyV-specific CD8^+^ T-cell responses [16]. The virus replicates within epithelial cells, causing direct cytopathic effects such as nuclear enlargement, chromatin margination, and cell lysis [17].

### 3.2. Immunopathogenesis of BK Virus Reactivation

In immunocompromised transplant recipients, BK virus reactivation produces two distinct but mechanistically related syndromes in the urinary tract. In allogeneic HCT, particularly with myeloablative conditioning or PTCy, conditioning-related epithelial injury and delayed immune reconstitution facilitate BKPyV replication in the bladder urothelium [18,19]. The viral large T antigen disrupts the p53 and Rb tumor suppressor pathways, leading to unchecked proliferation of infected cells, which eventually lyse and release viral antigens. The resulting epithelial denudation, exposure of submucosal vasculature, and influx of activated cytotoxic CD8^+^ T cells and NK cells—amplified during immune reconstitution—culminate in BKPyV-HC with hematuria, dysuria, and clot retention. In kidney transplant recipients (KTRs), BKPyV reactivation preferentially targets renal tubular epithelial cells, where similar mechanisms of viral replication and immune-mediated injury produce tubular damage, interstitial inflammation, and progressive fibrosis [20]. Histologically, this process is marked by viral inclusions, tubulitis, and interstitial fibrosis, and is a major contributor to allograft dysfunction and loss. Together, these syndromes illustrate how BKPyV-induced cytopathic injury and dysregulated host immune responses interact to drive organ-specific pathology in the urinary tract [21].

### 3.3. Immunopathological Mechanisms

The pathogenicity of BKPyV is not solely attributed to direct cytopathic damage. Urothelial inflammation is driven in part by host immune responses, especially delayed T-cell recovery or excessive activation during immune reconstitution [22]. Inflammatory cytokines such as IL-6, IFN-γ, and TNF-α further aggravate tissue injury [23].

### 3.4. Emerging Insights

Recent high-throughput sequencing and proteomic studies have uncovered BKPyV-mediated alterations in host cell cycle regulators such as pRb and p53, as well as interactions between viral proteins and DNA damage response pathways [24]. These findings highlight BKPyV’s oncogenic potential and its ability to persist in a subclinical replicative state [25].

## 4. Differences in Pathogenesis and Monitoring: BKPyVN vs. BKPyV-HC

### 4.1. Clinical Manifestations

The clinical presentation of BKPyVN significantly differs from that of BK-HC due to the involvement of various organ systems. BKPyVN is usually concealed and frequently shows no symptoms in its initial phases. The initial indication often observed is a persistent, unaccounted increase in serum creatinine, detected during routine monitoring after a kidney transplant. Patients generally do not display specific symptoms until significant renal impairment transpires, at which point they may experience reduced urine output, hypertension, or edema. It is not common to have flank pain. Failure to detect the virus early may permit sustained replication, contributing to immune dysregulation, fibrotic changes in the tubulointerstitium, and eventual graft dysfunction or failure.

In contrast, BKPyV-HC is clinically evident and primarily affects the lower urinary tract. Patients often have hematuria, which can range from microscopic to large and may be accompanied by clots. Other associated symptoms include dysuria, increased urinary frequency and urgency, as well as suprapubic pain. In severe cases, patients may experience clot retention, bladder outlet obstruction, and anemia resulting from persistent hemorrhage; infrequently, issues in the upper urinary system, such as hydronephrosis, may occur. BKPyV-HC generally presents 2 to 8 weeks following allogeneic HCT [26,27]; however, a delayed onset may transpire during immune reconstitution.

The different clinical profiles show while BKPyVN slowly becomes worse over time, BKPyV-HC has sudden and obvious symptoms [28]. This shows how important it is to keep an eye on nephropathy in the lab and to use a symptom-based approach to diagnosing BKPyV-HC. Disease severity is often stratified using a four-grade classification (Grade I–IV), with escalating morbidity in higher grades [29]. Established risk factors include myeloablative conditioning regimens (especially cyclophosphamide or total body irradiation), use of unrelated or haploidentical donors, male sex, older pediatric age, cytomegalovirus (CMV) co-infection, and acute GVHD [30,31,32].

### 4.2. Differential Diagnosis and Alternative Etiologies

Diagnosing BKPyV-associated nephropathy (BKPyVN) involves both laboratory surveillance and confirmatory histopathology. Early detection relies on regular plasma BKPyV DNA monitoring via PCR, with sustained viremia, particularly exceeding 10,000 copies/mL, closely linked to the onset of nephropathy [33]. Urinary BKPyV load may serve as an early indicator of reactivation, although asymptomatic viruria is common and not necessarily predictive of clinical disease.

In cases of persistent viremia and/or rising serum creatinine, a kidney biopsy remains the gold standard for diagnosis. Histopathology commonly reveals cytopathic effects in tubular epithelial cells, accompanied by interstitial inflammation and SV40-positive immunostaining, confirming BKPyV involvement and helping to distinguish BKPyVN from acute rejection or alternative causes of graft impairment [34].

Diagnosis of BKPyV-HC primarily relies on clinical evaluation and symptomatology. In HCT recipients, hematuria is a common clinical concern, often accompanied by symptoms such as dysuria, urinary urgency, and suprapubic discomfort. Elevated urinary BKPyV DNA levels are frequently observed; however, this finding is nonspecific, as viruria is also common among asymptomatic transplant patients. While plasma viral load may increase, its correlation with disease severity remains variable and inconsistent. Hematuria stands as the key diagnostic indicator within the relevant clinical context, whereas urine quantitative polymerase chain reaction (qPCR) acts as an additional measure rather than a conclusive one. Cystoscopy or imaging is generally reserved for severe cases to eliminate the possibility of blockage or other underlying conditions. Infections such as adenovirus or CMV are frequently assessed to rule out co-infection.

A differential diagnosis of hemorrhagic cystitis (HC) should include both non-infectious and infectious etiologies. Direct urothelial toxicity may result from chemotherapy drugs including cyclophosphamide, ifosfamide, and busulfan. Conditioning-related mucosal injury is also common, particularly after myeloablative regimens. Among infectious causes, adenovirus is a key viral mimic of BKPyV-HC, while CMV, HSV, and JC virus may also contribute. Bacterial and fungal pathogens, though rare, should be considered in neutropenic or catheterized patients. Distinguishing these causes is critical to guide appropriate management.

qPCR assays for BK virus DNA in urine and plasma remain the gold standard for diagnosis, with viruria levels exceeding 10^7^ copies/mL usually associated with symptomatic cases [35]. Viremia correlates with more severe disease and systemic involvement. While cystoscopy is not routinely necessary, it may be employed in cases of persistent bleeding or obstruction. Findings show diffuse mucosal hemorrhage and inflammation [36]. Notably, the detection of BK virus alone does not confirm BKPyV-HC, as asymptomatic viruria is common after HCT [7]. Clinical correlation remains crucial.

### 4.3. Treatment and Outcomes of BKPyV-HC

The outcome of BKPyVN depends on early detection and prompt management. Early reduction of immunosuppression remains the cornerstone of treatment and can stabilize or even restore allograft function. However, if diagnosis is delayed, ongoing viral replication and inflammation lead to chronic tubulointerstitial fibrosis and graft failure, with graft loss reported in up to 10–50% of biopsy-proven cases [37]. No antiviral therapy has been conclusively proven effective, but adjunctive agents such as leflunomide, cidofovir, intravenous immunoglobulin, and fluoroquinolones have been used with variable success [38]. Early data suggest that emerging therapies, particularly BK virus-specific T-cell transfer, may facilitate viral elimination without substantially increasing the risk of rejection [39]. Research is also focused on identifying predictive biomarkers and developing targeted antivirals to enable earlier, personalized interventions.

In contrast, the clinical course of BKPyV-HC is usually acute and self-limited but varies in severity. The natural history of BKPyV-HC varies by severity and patient-specific variables. Mild cases frequently resolve with supportive care (e.g., hydration and analgesics), while more severe cases may last 4–6 weeks and require intensive interventions such as bladder irrigation, transfusions, and temporary immunosuppression reduction.

Emerging therapies such as BKPyV-specific cytotoxic T lymphocytes (CTLs) and mesenchymal stromal cells (MSCs) have shown promise in early studies [40]. While the majority of patients recover without lasting complications, persistent BKPyV-HC can result in urinary tract infections, renal impairment, and prolonged hospitalization. It is infrequently fatal but contributes significantly to post-transplant morbidity [41].

Early identification of at-risk patients, longitudinal monitoring of viral loads, and early therapeutic strategies are critical to improve outcomes. Research efforts are ongoing to develop predictive biomarkers and targeted antiviral therapies [42]. BKPyV-HC is a major complication following allogeneic HCT, and the identification of reliable biomarkers is essential for early detection, risk stratification, and management. Among the most studied markers is plasma BK viral load, with thresholds ≥10^4^ copies/mL demonstrating strong predictive value for the development of BKPyV-HC, particularly in pediatric patients [43]. In contrast, urinary decoy cells—although commonly detected—exhibit lower specificity and sensitivity [44].

Emerging markers such as polyomavirus–Haufen aggregates in urine, initially validated for BKPyVN, may reflect direct pathogenic invasion by the virus and hold promise for detecting invasive urothelial disease, although their use in BKPyV-HC remains underexplored [45]. Additionally, genotype-specific neutralizing antibodies may help stratify host susceptibility and guide personalized monitoring strategies [46]. Investigational parameters, including transferrin saturation and iron metabolism markers, are being evaluated as adjunctive indicators of BKPyV reactivation risk [22]. Future research should aim to investigate the prognostic significance of these biomarkers in upcoming studies and integrate them into individualized, risk-adapted algorithms to improve outcomes in HCT recipients. Table 1 outlined the comparative features of the two entities.

## 5. Role of BKPyV Monitoring and Risk Stratification in HCT Recipients

BKPyV DNA quantification in urine and blood plays complementary roles in clinical diagnosis, patient risk profil-ing, and long-term follow-up of BKPyV-associated complications in HCT recipients. Early identification of high-risk individuals is essential, given the multifactorial pathogenesis and substantial morbidity associated with BKPyV-HC.

Urinary BKPyV quantitative PCR (qPCR) is a highly sensitive tool for detecting viral reactivation. High-level viruria—usually defined as >10^7^ copies/mL—is frequently observed prior to the clinical onset of BKPyV-HC [26,47]. However, urinary BKPyV shedding may occur in asymptomatic patients, limiting its specificity, especially during routine surveillance in the early post-transplant period [27].

In contrast, plasma or whole blood BKPyV DNA levels, measured by qPCR, are more specific for systemic viral replication and correlate with heightened disease severity, including BKPyVN and grade III–IV hemorrhagic cystitis [48,49]. A plasma viral load exceeding 10^4^ copies/mL is commonly used as a clinically meaningful threshold and may warrant therapeutic intervention [50].

Several risk factors increase the probability of BKPyV reactivation progressing to clinically significant disease:Transplant-related factors: myeloablative conditioning, haploidentical or unrelated donor HCT, and use of PTCy for GVHD prophylaxis have been consistently linked to a higher incidence of BKPyV-HC [8,9,10].Immunologic factors: delayed reconstitution of BKPyV-specific CD4+ and CD8+ T cells, as seen in patients with severe GVHD or sustained immunosuppression, compromises viral control [11,12].Patient-specific factors: male sex, advanced age, or a history of bladder injury (e.g., from prior chemotherapy or irradiation) may also predispose to BKPyV-HC [9,11].Virologic factors: high pre-transplant viruria, certain BKPyV genotypes, and rapid viral load kinetics have been associated with more aggressive disease courses [32].

## 6. Preventive Strategies for BKPyV-HC

BKPyV-HC remains a significant complication following allogeneic HCT, particularly in patients receiving myeloablative conditioning or PTCy [26,51,52]. Although common and linked to significant morbidity, there are currently no standardized prophylactic protocols. Current preventive approaches focus on reducing viral reactivation, mitigating urothelial injury, and tailoring care based on individualized risk assessment.

### 6.1. Supportive Measures and Monitoring Strategies

Preventive efforts for BKPyV-HC rely on early monitoring and supportive care. Individuals undergoing HCT may benefit from enhanced monitoring of BKPyV reactivation, particularly during periods of intense immunosuppression or mucosal vulnerability [53,54].

Supportive measures play a central role in minimizing bladder injury and mitigating the severity of HC. These include vigorous intravenous hydration and pharmacologically induced diuresis, which serve to dilute urinary toxins and reduce contact time between potentially harmful agents and the bladder urothelium.

Mesna, a well-established uroprotective agent, is routinely used to prevent chemical cystitis related to cyclophosphamide administration. While mesna does not directly inhibit BKPyV reactivation, it may help preserve urothelial integrity, potentially decreasing the bladder’s vulnerability to virus-induced injury [55]. Continued optimization of supportive strategies and standardized monitoring protocols may improve clinical outcomes in patients at risk for BKPyV-HC.

### 6.2. Pharmacologic and Antiviral Prophylaxis

Various antiviral agents have been investigated for prophylaxis, including cidofovir, leflunomide, and fluoroquinolones. Low-dose cidofovir has shown efficacy in reducing BKPyV viral loads in retrospective studies; however, its use is constrained by nephrotoxicity and the absence of strong prospective validation [56,57]. Fluoroquinolones, initially promising due to in vitro anti-BKPyV activity, have not demonstrated sustained therapeutic effect in randomized trials, and their prophylactic use is no longer routinely recommended [58].

Leflunomide, an immunomodulatory agent that inhibits pyrimidine synthesis, has been used in BKPyV infections with inconsistent efficacy. While some transplant recipients have responded favorably, its use in BKPyV-HC specifically remains under-investigated, with limited data indicating a complete response rate of approximately 50% [59]. Table 2 summarizes therapeutic strategies for BKPyVN and BKPyV-HC derived from clinical trials and observational studies.

### 6.3. Emerging and Investigational Therapies

While supportive care remains central, multiple adjunctive therapies targeting urothelial repair, antiviral activity, or immune reconstitution are being evaluated in refractory BKPyV-HC.

#### 6.3.1. Recombinant Growth Factors and Antivirals

##### Keratinocyte Growth Factor (KGF/Palifermin)

Through FGFR2b receptor activation, KGF promotes proliferation and migration of urothelial basal cells, accelerates re-epithelialization, and increases cytoprotective cytokine production. Pilot studies in HCT recipients demonstrated reduced mucosal injury severity; however, its prophylactic and therapeutic roles are limited by high cost and inconsistent phase II data [67].

##### Brincidofovir

Brincidofovir is a lipid-conjugated derivative of cidofovir with improved intracellular delivery and reduced nephrotoxicity. While in vitro data confirm potent inhibition of BKPyV DNA polymerase, clinical experience in BKPyV-HC remains anecdotal, and controlled trials are lacking [68].

#### 6.3.2. Hyperbaric Oxygen Therapy (HBOT)

HBOT has been applied as an adjunctive treatment for refractory BKPyV-HC. The proposed mechanisms include (1) improving oxygen delivery to hypoxic bladder mucocosa, which promotes epithelial proliferation and wound healing; (2) stimulation of angiogenesis through upregulation of vascular endothelial growth factor (VEGF); and (3) attenuation of local inflammation via downregulation of pro-inflammatory cytokine release [69].

In a retrospective series by Savva-Bordalo et al., 15 out of 16 patients (94%) with BKPYV-HC achieved complete resolution after a median of 13 HBOT sessions (range 4–84), accompanied by a marked reduction in urinary BKPyV viral load [69]. In another comparative study, all eight patients in the HBOT group achieved complete resolution versus 62.5% in the control group, with a median of 10 sessions (range 8–12) and a median time to resolution of 14.5 days (range 5–25 days). Treatment was generally well tolerated, with only occasional discontinuations due to barotrauma or claustrophobia.

These data support HBOT as a promising adjunctive option for patients who fail conventional antiviral and supportive measures; however, controlled prospective trials are still needed to confirm efficacy and to define standardized treatment parameters [70].

#### 6.3.3. Adoptive Cellular Therapies

##### BK Virus-Specific CTLs

These are generated from donor or third-party lymphocytes expanded ex vivo and in fused into recipients. These CTLs selectively target BKPyV-infected urothelial cells, restoring virus-specific immunity. In a prospective pilot study, 6 of 7 patients with severe BKPyV-HC achieved marked viral load reduction and symptomatic improvement without triggering significant GVHD [71,72].

##### Donor Lymphocyte Infusion (DLI)

Although DLIs can exert antiviral effects, they carry a substantial GVHD risk and are generally reserved for concurrent mixed chimerism.

##### Mesenchymal Stromal Cells (MSCs)

Umbilical cord-derived MSCs have dual immunosuppressive (regulating alloreactive T cells) and regenerative (stimulating tissue repair) properties. A pediatric prospective study involving 13 children with severe BKPyV-HC reported 100% complete clinical resolution, without increased GVHD incidence [73].

#### 6.3.4. Other Agents

##### Vidarabine

This purine nucleoside analogue interferes with DNA polymerase, but its role against BKPyV remains investigational, with limited efficacy signals from small case series.

##### Estrogen Therapy

Estrogens may protect bladder microvasculature and enhance mucosal integrity, but experience is limited, and the high cost restricts broad clinical use [74].

##### KGF Rescue Therapy

Although theoretically beneficial, late KGF administration during established HC has not demonstrated clear clinical efficacy, and its use is discouraged outside research protocols.

## 7. Intravesical Therapy for BKPyV-Associated Hemorrhagic Cystitis

Intravesical therapy offers a bladder-targeted approach to managing refractory BKPyV-HC, particularly in patients at risk for nephrotoxicity from systemic therapies. Agents such as cidofovir, fibrin glue, platelet-rich plasma (PRP), formalin, alum, and prostaglandins have been employed to control bleeding and preserve bladder function.

Intravesical cidofovir delivers the antiviral locally, potentially avoiding systemic nephrotoxicity. Although limited primarily to case reports and small series, intravesical administration has shown variable clinical and virologic responses in patients with severe BKPyV-HC [75,76]. Nonetheless, systemic absorption may still occur, and cases of renal dysfunction have been documented [75].

Fibrin glue, applied intravesically, has demonstrated favorable outcomes in managing refractory HC. One prospective study reported complete clinical response in 29 of 35 patients by day 14, with comparable overall survival to other allogeneic HCT recipients [77]. It provides a mechanical seal over bleeding urothelial surfaces, promoting mucosal healing.

Platelet-rich plasma (PRP), or autologous platelet gel, consists of concentrated platelets in plasma, devoid of leukocytes. Intravesical administration promotes hemostasis and urothelial repair by releasing growth factors such as PDGF and TGF-β; however, supporting evidence is limited to case reports and small-scale studies [78].

Formalin acts by denaturing mucosal proteins and sealing blood vessels but is associated with significant toxicity, including suprapubic pain, bladder fibrosis, and vesicoureteral reflux. It is primarily indicated in situations of refractory, life-threatening bleeding that does not improve with less toxic or standard treatments [79].

Alum (aluminum potassium sulfate) similarly induces protein precipitation, reducing capillary oozing and promoting mucosal hemostasis. While better tolerated than formalin, its use may be contraindicated in patients with renal impairment due to the risk of aluminum absorption [80].

Prostaglandin E2 (PGE2), though its use has diminished over time, has been reported to facilitate bleeding cessation through vasoconstriction and platelet aggregation. However, its use is largely historical, and evidence remains anecdotal [81].

## 8. Surgical Management of Refractory BKPyV-HC

When medical therapies fail, surgical interventions may be required to control life-threatening hemorrhage or address bladder damage.

Selective vesical artery embolization (SVAE) has been employed with success in managing refractory BKPyV-HC. Reported clinical response rates have reached as high as 80% [82]. However, complications such as bladder wall necrosis, gluteal muscle paresis, and perineal or skin necrosis have been observed, necessitating careful patient selection and post-procedural monitoring [83].

Supravesical urinary diversion, such as percutaneous nephrostomy or ileal conduit, may be considered to reduce bladder distension and urokinase exposure, thereby mitigating sustained injury to the urothelium. This approach may provide temporary bladder rest in severe cases [84].

Cystectomy remains a last resort for intractable BKPyV-HC unresponsive to medical and conservative surgical management. Although rarely performed, isolated case reports document symptom resolution and survival in patients undergoing cystectomy during profound immunosuppression and pancytopenia [85,86]. Given its invasiveness and high-risk profile, cystectomy should be reserved for cases with no alternative therapeutic options. Figure 1 depicts the pathophysiology and management algorithm of BKPYV-HC in HCT.

## 9. Current Guidelines and Consensus Statements

No universal guidelines exist for BKPyV-HC; management relies on expert consensus, focusing on supportive care, cautious immunosuppression reduction, and limited antiviral use. ECIL-6 emphasizes the importance of early detection, vigilant monitoring, and the consideration of emerging cellular therapies for treatment-refractory cases [87]. By contrast, BKPyVN guidelines (KDIGO, AST) recommend routine BK viral load screening, prompt immunosuppression adjustment, and individualized use of antivirals [88,89]. All consensus statements emphasize the urgent need for standardized protocols and clinical trials. Table 3 provides a comparative summary of current international guidelines addressing the diagnosis, monitoring, and management of BK polyomavirus-associated complications in HCT and KTR.

## 10. Therapeutic Advances

Recent developments in BKPyV research have focused on elucidating its pathogenesis and identifying novel therapeutic targets. Schneidewind et al. employed a three-dimensional (3D) urothelial cell culture model to mimic in vivo conditions more accurately, offering critical insights into the virus–host interaction during BKPyV infection [90]. This platform enables the study of the full BKPyV life cycle, including early gene expression, replication, and cell lysis, and holds promise for preclinical antiviral drug screening [91].

In parallel, signal transducer and activator of transcription 3 (STAT3) signaling has emerged as a key pathway exploited by BKPyV during infection. In a recent study, STAT3 activation was shown to upregulate interleukin-11 (IL-11) expression, contributing to viral persistence and urothelial injury. Notably, neutralizing antibodies against IL-11 demonstrated inhibitory effects on BKPyV replication in vitro, positioning IL-11 and STAT3 as potential immunotherapeutic targets in the management of BKPyV-associated disease [92].

The identification of such host-directed pathways provides a compelling rationale for further investigation into host–pathogen interactions, which may pave the way for non-cytotoxic, targeted antiviral therapies and adjunctive strategies to improve transplant-related outcomes.

## 11. Conclusions

In conclusion, the management of BKPyV-HC in allogeneic HCT remains a complex and evolving clinical challenge. Current strategies—primarily involving supportive care, virological monitoring, and tailored immunosuppression—provide only partial control. The clinical effectiveness of antiviral and cellular therapies remains constrained by the absence of robust, large-scale, prospective evidence.

Emerging modalities such as virus-specific T-cell therapy, targeted molecular agents, and regenerative strategies including MSC infusions represent promising avenues, yet remain to be thoroughly validated. Moving forward, the identification of predictive biomarkers and the development of individualized, preemptive treatment algorithms will be essential to reducing disease burden, minimizing complications, and improving long-term transplant outcomes.

## Figures and Tables

**Figure 1 viruses-17-01256-f001:**
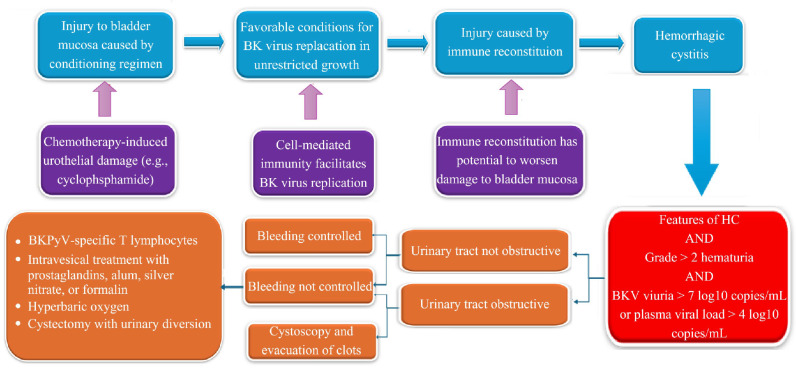
Schematic representation of the pathogenesis and stepwise management of BK polyomavirus-associated hemorrhagic cystitis (BKPyV-HC) following hematopoietic cell transplantation. Conditioning-related urothelial injury facilitates BKPyV replication, which is compounded by the effects of immune reconstitution. Diagnosis is based on hematuria grade ≥2 with BKPyV viruria >10^7^ copies/mL or plasma viral load >10^4^ copies/mL. Management is guided by bleeding control and urinary tract obstruction, ranging from supportive care to intravesical therapies, hyperbaric oxygen, virus-specific T cells, or surgical intervention. The figure is not owned by the authors. All credits go to its rightful owner.

**Table 1 viruses-17-01256-t001:** Comparative features of BK virus-associated nephropathy (BKPyVN) and BK virus-associated hemorrhagic cystitis (BKPyV-HC).

Feature	BKPyVN	BKPyV-HC
Primary patient population	Kidney transplant	Allogeneic HCT
Primary organ affected	Renal allograft (tubules/interstitium)	Bladder urothelium
Trigger for pathogenesis	Immunosuppression leading to viral replication	Urothelial injury from conditioning + viral reactivation
Clinical manifestation	Gradual rise in serum creatinine, renal dysfunction	Hematuria (ranging from microscopic to gross), dysuria, clot retention
Key timing	Usually >3 months post-kidney transplant	2–8 weeks post-HCT (can be late-onset)
Monitoring focus	Plasma BK viral load and kidney function (routine screening)	Clinical signs and symptoms; BK viral load less predictive
Gold standard for diagnosis	Kidney biopsy	Clinical + high urine BK viral load; biopsy rarely needed

Abbreviations: HCT—hematopoietic cell transplantation; BKPyVN—BK polyomavirus-associated nephropathy; BKPyV-HC—BK polyomavirus-associated hemorrhagic cystitis.

**Table 2 viruses-17-01256-t002:** Therapeutic strategies for BKPyVN and BKPyV-HC: insights from clinical trials and studies.

Agent/Strategy	Study Type and Size	Key Outcomes	Toxicity/Safety Findings	Evidence Level and Limitations	References
Cidofovir (IV)	Retrospective cohort (*n* = 27 HCT recipients with BKPyV-HC)	Complete response in 81.5%; partial response in 7.4%	~30% developed renal impairment; ↓ creatinine clearance by 27%	Observational; no RCTs; single-dose protocol	[56]
Cidofovir (IV)	Retrospective series (*n* = 12 transplant recipients)	Significant reduction in HC severity	Not reported	Small sample; uncontrolled design	[60]
Cidofovir ± Intravesical	Single-center case series (*n* = 27 allo-HCT patients)	60–100% of CRs observed independently of the dose or administration route.	The main toxicity reported was renal failure	Very small sample; emphasizes safety/dosing concerns	[52]
Cidofovir vs. Supportive Care	Ongoing RCT (NCT01295645)	Results pending	—	Awaiting results; randomized design promising	[61]
Leflunomide	Systematic review (12 studies; 267 KTR with BKPyVN)	BK viremia clearance ranged from 33–92%; ~10% graft loss	Hemolysis, thrombotic microangiopathy	Heterogeneous study protocols; risk of bias	[62]
Leflunomide + Everolimus	Case reports/series (*n* = 4–26 patients)	Reported viral clearance; stable graft function	High dose (>40 µg/mL) associated with hemolysis	No control group; variable dosing protocols	[63]
Leflunomide (Pediatrics)	Pediatric KTR case series (*n* ≈ unknown); ECTR-X 2023 abstract	All cleared viremia (~3.4 months); renal function preserved	No hepatotoxicity or anemia reported	Retrospective; limited cohort size	[64]
IVIG ± Leflunomide	RCT (*n* = 16 adult KTR with BKPyVN)	7/8 cleared viremia in combo group vs. 3/7 with IVIG alone	Not reported	Small sample size; short-term follow-up	[65]
Brincidofovir	No BKPyV-specific trials; negative phase III in CMV/adenovirus	—	Increased gastrointestinal and severe adverse events	No data for BKPyV; clinical development discontinued	[66]

Abbreviations: HCT—hematopoietic cell transplantation; KTR—kidney transplant recipient; BKPyVN—BK polyomavirus-associated nephropathy; CR—complete response; IVIG—intravenous immunoglobulin; RCT—randomized controlled trial.

**Table 3 viruses-17-01256-t003:** Summary of key guidelines and consensus statements for BK virus–associated complications.

Guideline/Statement	Target Population	Key Recommendations	Evidence Level/Notes	References
ECIL-6 (2015)	HCT recipients (focus on BKPyV-HC)	Routine screening of BK viruria/viremia in high-risk patients Supportive care: hydration, analgesics, bladder irrigation Consider immunosuppression reduction if feasible Antivirals (e.g., cidofovir) only for refractory/severe cases Investigational use of virus-specific cytotoxic T lymphocytes in resistant cases	Based on expert consensus; no randomized controlled trials; emphasizes early detection and prevention	[87]
AST Infectious Diseases Community of Practice (2019)	KTR (BKPyVN)	Plasma BK viral load screening monthly for 3–6 months, then periodically Stepwise immunosuppression reduction as first-line therapy Consider leflunomide, cidofovir, or conversion to mTOR inhibitors if viremia persists No routine antiviral prophylaxis recommended	Recommendations derived from observational studies and expert opinion	[88]
KDIGO Transplant Guidelines (2020)	KTR (BKPyVN)	Screen for BK viremia at least monthly for the first 3–6 months, then every 3 months until the end of year 1, and with any unexplained allograft dysfunction Prompt reduction of immunosuppression with sustained viremia (>10,000 copies/mL) Adjunctive antivirals considered in refractory cases	Evidence-based guideline; emphasizes screening and early intervention	[89]

Abbreviations: AST—American Society of Transplantation; HCT—hematopoietic cell transplantation; BKPyVN—BK polyomavirus-associated nephropathy; BKPyV-HC—BK polyomavirus-associated hemorrhagic cystitis; ECIL-6—the 6th European Conference on Infections in Leukemia; KDIGO—Kidney Disease: Improving Global Outcomes; KTR—kidney transplant recipient.

## Data Availability

Not applicable.

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
