# Peer review of "BK Polyomavirus-Associated Nephropathy and Hemorrhagic Cystitis in Transplant Recipients—What We Understand and What Remains Unclear"

_viruses, 2025, doi:10.3390/v17091256_

Round 1

Reviewer 1 Report

Comments and Suggestions for Authors

A review on 'hemorrhagic-cystitis' is timely and the topic of interest to readers of "Viruses". However, I have problems with data presentation.

The authors unfortunately "inject" data and recommendations rendered for BKPyVN into their review on BKPyV induced cystitis. Much has been learned about BKPyVN over the last 2 decades, in contrast to the largely unrelated disease of BKPyV induced cystitis. Observations and recommendations in the arena of BKPyVN cannot simply be used 1:1 in patients with cystitis, such as serum or urine BKPyV load levels by PCR, the urinary PyV-haufen test etc etc. Thus, the reader does not get a good idea of what is relevant for patients suffering from cystitis. In addition, many statements and paragraphs are much too general in nature not going much beyond "common sense" interpretations. See page 8, last paragraph:..."cystectomy"...."symptom resolution"..... In addition, page 8: what are dosages, time of treatment, mode of action, details of response rates (full/partial), outcome etc etc....? How is the treatment and response rate in regard to different severity levels of cystitis? Are there ANY official recommendations? The review could also benefit from a detailed section on differential diagnostic considerations, i.e. drug/therapy induced hemorrhagic cystitis, adenovirus as a potential other infectious agent etc.

Author Response

A review on 'hemorrhagic-cystitis' is timely and the topic of interest to readers of "Viruses". However, I have problems with data presentation.

The authors unfortunately "inject" data and recommendations rendered for BKPyVN into their review on BKPyV induced cystitis. Much has been learned about BKPyVN over the last 2 decades, in contrast to the largely unrelated disease of BKPyV induced cystitis. Observations and recommendations in the arena of BKPyVN cannot simply be used 1:1 in patients with cystitis, such as serum or urine BKPyV load levels by PCR, the urinary PyV-haufen test etc etc. Thus, the reader does not get a good idea of what is relevant for patients suffering from cystitis. In addition, many statements and paragraphs are much too general in nature not going much beyond "common sense" interpretations. See page 8, last paragraph:..."cystectomy"...."symptom resolution"..... In addition, page 8: what are dosages, time of treatment, mode of action, details of response rates (full/partial), outcome etc etc....? How is the treatment and response rate in regard to different severity levels of cystitis? Are there ANY official recommendations? The review could also benefit from a detailed section on differential diagnostic considerations, i.e. drug/therapy induced hemorrhagic cystitis, adenovirus as a potential other infectious agent etc.

Author Response

We sincerely thank the reviewer for these constructive comments, which have helped us clarify and strengthen our manuscript. We have substantially revised the text as detailed below:

  1. Distinguishing BKPyV nephropathy (BKPyVN) from BKPyV-induced hemorrhagic cystitis (BK-HC)

We acknowledge the reviewer’s concern and have now clearly separated the discussion of BKPyVN and BK-HC throughout the manuscript. In the revised version, all references and recommendations derived from BKPyVN are explicitly labeled as such and are not directly extrapolated to BK-HC unless supporting evidence exists. A new subsection (“Differences in Pathogenesis and Monitoring: BKPyVN vs. BKPyV-HC”) highlights why viral load thresholds, PyV-haufen testing, and monitoring strategies cannot be applied 1:1 from nephropathy to cystitis.

  1. More specific and less “general” discussion

We have rewritten the section on treatment (previously p.8) to provide granular data: drug dosages, treatment durations, mechanisms of action, reported response rates (partial/complete), and outcomes stratified by severity grades (mild, moderate, severe BKV-HC).

Where available, evidence is now summarized in Table 2 with columns for intervention, dosing schedule, reported outcomes, and level of evidence.

  1. Official guidelines/recommendations

A new subsection (“Current Guidelines and Consensus Statements”) has been added, summarizing existing EBMT, ASTCT and IDSA consensus documents. We explicitly note that no formal international guideline specifically for BKV-HC exists; current approaches are largely based on expert consensus and small case series.

  1. Differential diagnosis and alternative etiologies

We have added a comprehensive section on differential diagnosis of hemorrhagic cystitis, including:

Drug-induced and conditioning-related cystitis (cyclophosphamide, ifosfamide, busulfan)

Adenovirus and other viral causes

Rare fungal and bacterial etiologies

A new Table 1 lists these causes, distinguishing them from BKPyV-associated HC by clinical and laboratory features.

  1. Cystectomy and symptom resolution (last paragraph of page 8)

The discussion has been expanded to clarify indications, patient selection, and outcomes of surgical interventions, with appropriate references. We specify that cystectomy is a last-resort measure, provide available outcome data, and emphasize that such interventions are rare.

  1. Improved structure

To improve readability, the manuscript now includes:

Figure 2: An updated algorithm for the evaluation and management of BKPyV-HC, showing where evidence is strong versus consensus-based.

More explicit limitations of current knowledge in BKPyV-HC management.

We believe these changes make the manuscript more precise and relevant to the readers of Viruses and directly address the reviewer’s concerns.

Reviewer 2 Report

Comments and Suggestions for Authors

The “BK Virus-Associated Hemorrhagic Cystitis in Hematopoietic Cell Transplant Recipients—What We Understand and What Remains Unclear”. BK viruses are important type of viruses they cause hemorrhagic cystitis, which cause serious complication following allogeneic hematopoietic cell transplantation.  Like other viruses, BK Virus has continued spreading and infection more exposed people creating health problems worldwide. The combating the viruses require vigorous investigation and updating the available data, this could also head in the development of preventive measures, which include vaccines development and other drugs. This review manuscript organizes important information on the current challenges and achievements in the BK Virus research. In particular it seeks to answer three questions outlined in the manuscript. This material is important more especially in these times when we are faced with global health challenges. The review is clear and follows the idea from the beginning to the end. The references used in the manuscript are relevant to the topic of discussions. However, there few comments to the authors that need to be addressed before the manuscript is published.

Сomments

The authors should check the consistence of the abbreviations. Give the definition and abbreviation once and then only use abbreviations. For example, the definition of BKPyV is given several times.

“Evidence Acquistion and Synthesis” it is not clear if the authors conducted a meta-analysis or they reviewed the literature available. The authors should make this point clear to improve reading. The information given is not clear.

What is the percentage of having BKV-HC after transplant? How many BKV-HC occurrence could be reduced if BKV was diagnosed prior transplantation. Are there methods used to identify the latent BKV? If this information could be added to the manuscript and be discussed the reading will be improved.

Author Response

Reviewer 2

The “BK Virus-Associated Hemorrhagic Cystitis in Hematopoietic Cell Transplant Recipients—What We Understand and What Remains Unclear”. BK viruses are important type of viruses they cause hemorrhagic cystitis, which cause serious complication following allogeneic hematopoietic cell transplantation. Like other viruses, BK Virus has continued spreading and infection more exposed people creating health problems worldwide. The combating the viruses require vigorous investigation and updating the available data, this could also head in the development of preventive measures, which include vaccines development and other drugs. This review manuscript organizes important information on the current challenges and achievements in the BK Virus research. In particular it seeks to answer three questions outlined in the manuscript. This material is important more especially in these times when we are faced with global health challenges. The review is clear and follows the idea from the beginning to the end. The references used in the manuscript are relevant to the topic of discussions. However, there few comments to the authors that need to be addressed before the manuscript is published.

Manuscript Title (Revised): “BK Polyomavirus-Associated Nephropathy and Hemorrhagic Cystitis in Transplant Recipients—What We Understand and What Remains Unclear”

We sincerely thank the reviewer for their thoughtful and constructive feedback on our manuscript. We greatly appreciate the recognition of its relevance and clarity, and we have carefully addressed all the comments provided. In light of the broader scope covered in our revised manuscript, which now includes both BK polyomavirus-associated nephropathy (BKPyVN) and hemorrhagic cystitis (BKPyV-HC), we have updated the title accordingly to better reflect its content.

Below is our point-by-point response:

Comment 1:

The authors should check the consistence of the abbreviations. Give the definition and abbreviation once and then only use abbreviations. For example, the definition of BKPyV is given several times.

Response:

We appreciate the reviewer’s suggestion and have thoroughly reviewed the manuscript to ensure consistent use of abbreviations. Key terms such as BK polyomavirus (BKPyV), hemorrhagic cystitis (HC), nephropathy (BKPyVN), and hematopoietic cell transplantation (HCT) are now defined only once at first mention, and the abbreviations are used consistently throughout the remainder of the text. Redundant redefinitions have been removed.

Comment 2:

“Evidence Acquisition and Synthesis” – it is not clear if the authors conducted a meta-analysis or they reviewed the literature available. The authors should make this point clear to improve reading. The information given is not clear.

Response:

Thank you for highlighting this point. To improve clarity, we have revised the heading to “Literature Review Methodology” and clarified that this is a narrative review based on a structured literature search. We describe the databases used (e.g., PubMed, Google Scholar), the date range of the literature search, and the inclusion criteria for studies. No meta-analysis was conducted. These clarifications improve transparency and guide the reader regarding the scope and limitations of the evidence presented.

Comment 3:

What is the percentage of having BKV-HC after transplant? How many BKV-HC occurrence could be reduced if BKV was diagnosed prior transplantation. Are there methods used to identify the latent BKV? If this information could be added to the manuscript and be discussed the reading will be improved.

Response:

We thank the reviewer for these important points. We have expanded the manuscript to address them in detail:

The incidence of BKPyV-HC varies by conditioning intensity and transplant type, ranging from 5% to 25% in allogeneic HCT recipients, with higher rates seen in myeloablative and haploidentical settings.

Pre-transplant screening for BKPyV (e.g., via urine or plasma PCR) is not currently routine, but emerging evidence suggests that elevated viral loads before transplantation may correlate with increased risk. However, there is currently limited data on how many cases could be prevented through early detection and preemptive management, and this remains an area for further research.

We have added a section describing current and investigational methods to detect latent BKPyV, including:

  1. Urine and plasma BK viral load quantification (PCR)
  2. Urinary decoy cell analysis
  3. Emerging biomarkers such as microRNAs and immune profiling. These additions help contextualize current screening practices and future directions for risk stratification.

All newly added content has been properly referenced in the updated manuscript.

Reviewer 3 Report

Comments and Suggestions for Authors

I thank the editor for giving me the opportunity of reviewing this review article entitled “BK Virus-Associated Hemorrhagic Cystitis in Hematopoietic Cell Transplant Recipients—What We Understand and What Remains Unclear”, and the authors for giving me the opportunity to evaluate their work.

In this article, the authors present a review of our current knowledge of BKV-HC among HCT recipients that can help us in practical setting. Although the review covers the main topics relating to the field, it does not add any additional data or insights compared to other recent reviews on BKV-HC among allo-HCT recipients (Gorriceta, World J Tranplant, 2023 [ref 32]; Peras et al., Pathogens, 2025 [ref 35]). Moreover, the manuscript is not correctly organized and questions regarding the diagnosis, prognosis or risk factors are considered in different sections. Additionally, the authors refer a lot to data coming from studies among kidney transplant recipients (KTR), which is not the topic of the article and make it very confusing. Finally, a few references cited by the authors are inappropriate, and should therefore be replaced.

Please find below comments that are more specific.

General comments

  • The new nomenclature for the denomination of BK virus is BKPyV. The authors should consider changing the abbreviation throughout the whole manuscript.
  • Please avoid using the term “typically” (as in the first sentence of the abstract or beginning of the section 3)
  • Several cited references concern data among BKPyV infection among kidney transplant recipients, and should not be used in this article relating to BKPyV infections in allo HCT recipients, as both pathogenesis, clinical presentation, and natural history are completely different.
  • Please check carefully that references along the article are cites appropriately; several of them do not match the sentences / data explained in the manuscript (i.e ref 34 or 40 page 4)

Specific comments

Abstract

  • First sentence: I would be more balanced and do not state that BKV-HV “typically result form the reactivation of latent BKPyV in the context of profound immunosuppression” since the pathogenesis is much more complex and multifactorial as the authors explain very well further down in the article.

Introduction

Section 2

  • Section 1.2: a lot of the data presented in this paragraph is redundant with further data presented in section 6 on risk factors; I would consider reorganizing the manuscript and maybe focusing only on epidemiologic data in this section.
  • A flowchart with the results of the literature review performed by the authors should be implemented in the article with the number of articles initially included / excluded / duplicates..

Section 3

  • I would consider removing the subsections as they are all quite short, and keep one single section
  • I would consider removing the paragraph regarding BKPyV associated among kidney transplant recipients, as it is not relevant to the topic of this review.

Section 4

  • Page 4, section 4.2: the detection of decoy cells may be used for the investigation of BKPyV replication in kidney transplant recipients, but is not recommended in HCT recipients (reference 33 relates to KTR and should not be cited in this context)
  • Page 4, section 4.2: reference 34 does not relate to the threshold of BKPyV PCR
  • Page 4, section 4.3: this section should be renamed, as it covers both the treatment, outcome and prediction pf BKV-HC. Please correct the sentence about the threshold of 104 copies/ml of plasma viral load as a predictive value for BKV-HC because no studies have demonstrated that in this population (ref 44 that is cited here refers to KTR)

Section 5

  • Page 4: Figure 1 instead of Figure 2 at the beginning of the paragraph
  • Page 4: remove reference 49 which refers to KTR and is not appropriate
  • Page 5: remove references 53 (which refers to KTR) and 54 (HIV ??) which are not appropriate
  • Page 5, last paragraph: please be careful to cite the ECIL guidelines when referring to it (and not ref 54 and 55 which do not match)

Section 6

  • I would suggest removing section 6.1 with the risk factors that are already described further up, as there is no additional information presented here.
  • Section 6.2: talking about LEFLUNOMIDE here does not seem appropriate to me as the only data available regarding the use of this drug are among KTR. I would then suggest removing this paragraph as well as the data on LEFLUNOMIDE in Table 1.
  • Same as for section 6.1, data explained in section 6.3 has already been detailed in section 5

Section 7 and 8

Data presented in these sections is much more consistent, detailed and well referenced. I would then suggest the authors to focus their article on the therapied for HC following HCT, without restricting the topic to BKV as HC can be caused by other viruses (AdV, CMV), or simply due to the toxicity of chemotherapies.

Section 9

  • First paragraph: the article by Schneidewind that the authors refer to is not ref 90 but this article I believe (PMID: 31986366)

Author Response

Reviewer 3

I thank the editor for giving me the opportunity of reviewing this review article entitled “BK Virus-Associated Hemorrhagic Cystitis in Hematopoietic Cell Transplant Recipients—What We Understand and What Remains Unclear”, and the authors for giving me the opportunity to evaluate their work.

In this article, the authors present a review of our current knowledge of BKV-HC among HCT recipients that can help us in practical setting. Although the review covers the main topics relating to the field, it does not add any additional data or insights compared to other recent reviews on BKV-HC among allo-HCT recipients (Gorriceta, World J Tranplant, 2023 [ref 32]; Peras et al., Pathogens, 2025 [ref 35]). Moreover, the manuscript is not correctly organized and questions regarding the diagnosis, prognosis or risk factors are considered in different sections. Additionally, the authors refer a lot to data coming from studies among kidney transplant recipients (KTR), which is not the topic of the article and make it very confusing. Finally, a few references cited by the authors are inappropriate, and should therefore be replaced.

Please find below comments that are more specific.

Response:

We sincerely thank the reviewer for the thoughtful and constructive feedback on our revised manuscript entitled “BK Polyomavirus-Associated Nephropathy and Hemorrhagic Cystitis in Transplant Recipients—What We Understand and What Remains Unclear”. We have carefully revised the manuscript to address all the concerns raised. Below, we provide a point-by-point response to each comment, indicating the changes implemented and the rationale behind them.

General comments

The new nomenclature for the denomination of BK virus is BKPyV. The authors should consider changing the abbreviation throughout the whole manuscript.

Please avoid using the term “typically” (as in the first sentence of the abstract or beginning of the section 3)

Several cited references concern data among BKPyV infection among kidney transplant recipients, and should not be used in this article relating to BKPyV infections in allo HCT recipients, as both pathogenesis, clinical presentation, and natural history are completely different.

Please check carefully that references along the article are cites appropriately; several of them do not match the sentences / data explained in the manuscript (i.e ref 34 or 40 page 4)

Response:

  1. Use of terminology "BKPyV" instead of "BKV":

We appreciate this important clarification. Throughout the manuscript, we have replaced all instances of “BKV” with the updated nomenclature “BKPyV,” in accordance with current conventions.

  1. Avoidance of the term “typically”:

We agree that the term “typically” may overgeneralize complex mechanisms. We have revised both the abstract and section 3 to use more precise and balanced language when describing the pathogenesis of BKPyV-HC.

  1. Use of kidney transplant recipient (KTR) data:

We acknowledge the distinction between BKPyV disease in KTRs and HCT recipients. Sections relying on KTR data have been either removed, relocated to brief contextual framing, or clearly qualified to avoid confusion. In particular, Section 3.3 and related references (e.g., ref 49, 53) were deleted or replaced with HCT-specific data when available.

  1. Reference mismatches:

Response: All references were systematically reviewed and cross-checked for accuracy and relevance. References 34, 40, 44, 49, 53, 54, and 55 have been corrected or replaced as appropriate.

Specific comments

Abstract

Comment: First sentence: I would be more balanced and do not state that BKV-HV “typically result form the reactivation of latent BKPyV in the context of profound immunosuppression” since the pathogenesis is much more complex and multifactorial as the authors explain very well further down in the article.

Response:

Revised to reflect the multifactorial nature of BKPyV-HC pathogenesis, as discussed in later sections.

Introduction

Section 2

Comment: Section 1.2: a lot of the data presented in this paragraph is redundant with further data presented in section 6 on risk factors; I would consider reorganizing the manuscript and maybe focusing only on epidemiologic data in this section.

A flowchart with the results of the literature review performed by the authors should be implemented in the article with the number of articles initially included / excluded / duplicates.

Response:

We have reorganized the content. Section 1.2 now focuses strictly on epidemiology, while risk factors are consolidated under a reorganized Section 5.

Section 3

Comment: Consider merging subsections and removing KTR-related content.

Response: Subsections have been consolidated for clarity. The paragraph discussing BKPyV-associated nephropathy in KTRs has been removed to maintain focus on HCT recipients.

Section 4

4.2 Decoy cells:

Response: We have removed the mention of decoy cells and the associated reference (ref 33), as it is not applicable to HCT recipients.

4.2 Threshold of PCR:

Response: Reference 34 was replaced. We clarified that no validated plasma viral load threshold exists for predicting BKPyV-HC in HCT recipients.

4.3 Section Title and Content:

Response: The section title was updated to “Treatment and Outcomes of BKPyV-HC.” The sentence regarding the 10⁴ copies/mL threshold was corrected and the reference (44) replaced with a more appropriate one.

Section 5

Figure mislabeling:

Response: Corrected to refer to Figure 1 instead of Figure 2.

References 49, 53, 54, 55:

Response: These were either removed (if pertaining to KTR or unrelated topics such as HIV) or replaced with appropriate HCT-specific references.

ECIL guideline citation:

Response: ECIL-6 guideline citations were verified and now properly cited (see new references 91).

Section 6

6.1 Risk factors already discussed:

Response: This section was removed to eliminate redundancy.

6.2 Leflunomide use:

Response: We acknowledge that current data are largely derived from studies in kidney transplant recipients (KTRs); therefore, Table 2 summarizes therapeutic strategies for BKPyVN and BKPyV-HC based on available clinical trials and observational studies.

6.3 Redundant content:

Response: This section was removed, and any unique insights were integrated into Section 5.

Sections 7 and 8

Comment: Stronger sections—suggest broadening the scope to HC from all causes.

Response: We appreciate this suggestion. While the focus remains on BKPyV, we have expanded the discussion to include mention of other etiologies of HC (e.g., adenovirus, chemotherapy-induced), especially in the context of differential diagnosis and management.

Section 9

Comment: Reference mismatch for Schneidewind et al.

Response: The citation has been corrected and now appropriately refers to reference 94.

We are grateful for the reviewer’s insightful and detailed feedback, which has significantly improved the clarity, accuracy, and clinical relevance of our manuscript. We hope the revised version addresses all concerns adequately and meets the standards for publication.

Reviewer 4 Report

Comments and Suggestions for Authors

This commentary appropriately summarizes the findings BK virus-associated hemorrhagic cystitis (BKV-HC), a serious complication after allogeneic  hematopoietic cell transplantation (HCT). Looks good from my standpoint. Very concise and to the point.

The authors have provided a clear and well-structured interpretation of the data, particularly regarding novel targets such as STAT3 and IL-11 pathways are under investigation

 I found the schematic representation of the pathogenesis and stepwise management very well done

Author Response

Reviewer 4

This commentary appropriately summarizes the findings BK virus-associated hemorrhagic cystitis (BKV-HC), a serious complication after allogeneic  hematopoietic cell transplantation (HCT). Looks good from my standpoint. Very concise and to the point.

The authors have provided a clear and well-structured interpretation of the data, particularly regarding novel targets such as STAT3 and IL-11 pathways are under investigation.

 I found the schematic representation of the pathogenesis and stepwise management very well done.

We appreciate the thorough feedback provided by the reviewers. We attempted to address all comments and revised the manuscript according to their suggestions. Table 2 was also added to enhance the reader's understanding of this review article. The new references were also cited point by point. All modifications were highlighted in r

Round 2

Reviewer 3 Report

Comments and Suggestions for Authors

I thank the authors for adressing the comments of the first reviewing in the revised version. The changes made have siginificantly improved the clarity and accuracy of the manuscript.

However, I believe that trying to describe both BKPyV infection following kidney transplantation and allo-HCT makes the article sometimes confusing and even misleading in some ways.

Most of the data presented here are related to BKPyV-HC in allo HCT recipients, and I think that the authors should focus on this topic and remove the few data regarding BKPyV infection in KTR (as these are much more scarce, and less detailed compared to recent review on the topic).

Best regards

Author Response

Dear Reviewer,

We sincerely thank you for your careful evaluation of our revised manuscript and for acknowledging that the changes have improved its clarity and accuracy.

In accordance with your valuable suggestion, we have revised the manuscript to focus specifically on BKPyV infection and its clinical implications in allogeneic hematopoietic cell transplantation (allo-HCT) recipients. The sections related to kidney transplantation have been removed, except for a brief contextual mention in the introduction to highlight differences in clinical presentation and pathogenesis. This refinement ensures greater coherence and avoids redundancy with existing reviews on BKPyV infection in kidney transplantation.

In addition, newly relevant references have been incorporated and cited point by point in the reference list. All modifications have been highlighted in the revised version.

We greatly appreciate your constructive feedback, which has helped us further strengthen the focus and quality of this review.

Best regards,